Neck biomechanics indicate that giant Transylvanian azhdarchid pterosaurs were short-necked arch predators

Naish Darren eotyrannus@gmail.com 1
Witton Mark P. 2
1 Ocean and Earth Science, University of Southampton , Southampton , UK
2 School of Earth & Environmental Sciences, University of Portsmouth , Portsmouth , United Kingdom
Wedel Mathew
Electronic publication date: 2017 Jan 18
Publication date: 2017
Volume: 5
Electronic Location ID: e2908
Received 2016 Aug 19; Accepted 2016 Dec 13
Copyright: ©2017 Naish and Witton
Copyright year: 2017
Copyright holder: Naish and Witton
License: This is an open access article distributed under the terms of the Creative Commons Attribution License, which permits unrestricted use, distribution, reproduction and adaptation in any medium and for any purpose provided that it is properly attributed. For attribution, the original author(s), title, publication source (PeerJ) and either DOI or URL of the article must be cited.
License URL: https://creativecommons.org/licenses/by/4.0/

Keywords: Pterosaurs, Azhdarchids, Cretaceous, Biomechanics, Maastrichtian

Funding: The authors received no funding for this work.

==============================
Azhdarchid pterosaurs include the largest animals to ever take to the skies with some species exceeding 10 metres in wingspan and 220 kg in mass. Associated skeletons show that azhdarchids were long-necked, long-jawed predators that combined a wing planform suited for soaring with limb adaptations indicative of quadrupedal terrestrial foraging. The postcranial proportions of the group have been regarded as uniform overall, irrespective of their overall size, notwithstanding suggestions that minor variation may have been present. Here, we discuss a recently discovered giant azhdarchid neck vertebra referable to Hatzegopteryx from the Maastrichtian Sebeş Formation of the Transylvanian Basin, Romania, which shows how some azhdarchids departed markedly from conventional views on their proportions. This vertebra, which we consider a cervical VII, is 240 mm long as preserved and almost as wide. Among azhdarchid cervicals, it is remarkable for the thickness of its cortex (4–6 mm along its ventral wall) and robust proportions. By comparing its dimensions to other giant azhdarchid cervicals and to the more completely known necks of smaller taxa, we argue that Hatzegopteryx had a proportionally short, stocky neck highly resistant to torsion and compression. This specimen is one of several hinting at greater disparity within Azhdarchidae than previously considered, but is the first to demonstrate such proportional differences within giant taxa. On the assumption that other aspects of Hatzegopteryx functional anatomy were similar to those of other azhdarchids, and with reference to the absence of large terrestrial predators in the Maastrichtian of Transylvania, we suggest that this pterosaur played a dominant predatory role among the unusual palaeofauna of ancient Haţeg.

Introduction

Substantial recent interest in the largest known azhdarchid pterosaurs—the Upper Cretaceous taxa Arambourgiania philadelphiae, Quetzalcoatlus northropi and Hatzegopteryx thambema—has shed much light on their morphology, palaeoecology, and flight capabilities (Witton & Naish, 2008; Witton & Naish, 2015; Witton & Habib, 2010; Habib, 2013). This advanced pterodactyloid clade, deeply nested with the morphologically diverse Azhdarchoidea (Nessov, 1984; Kellner, 2003; Unwin, 2003; Andres & Myers, 2013), is noted for the proportionally elongate, edentulous jaws, remarkably long, cylindrical neck vertebrae and often unusually large size of its constituent taxa (Witton & Naish, 2008; Witton & Naish, 2015). Although azhdarchids are comparably well represented in the fossil record compared to other pterosaur groups, frustratingly little is known of their skeletal anatomy. Partial and complete skeletons of azhdarchids are known and are adequately documented (e.g., Godfrey & Currie, 2005), but the best represented taxa—Zhejiangopterus linhaiensis and Quetzalcoatlus sp.—remain only preliminarily described (Cai & Wei, 1993; Lawson, 1975; Kellner & Langston Jr, 1996). Hypotheses about flight, body mass, functional morphology, ecology and lifestyle, all of which remain controversial, are based predominantly on knowledge of inadequately described taxa (Witton & Naish, 2008; Witton & Naish, 2015; Averianov, 2013). Despite these shortfalls of evidence, azhdarchids have been widely assumed as uniform in anatomy and ecology (Unwin, 2005; Witton & Naish, 2008; Witton, 2013).

Azhdarchids are primarily characterised by their elongate, often tubular neck vertebrae (Nessov, 1984; Kellner, 2003; Unwin, 2003; Andres & Myers, 2013), and it is a familiar fact of the pterosaur literature that these often isolated fossils make up a substantial portion of their fossil record. That the giant azhdarchids had the same long necks as their smaller relatives has been verified by the discovery of several gigantic vertebrae, including University of Jordan, Department of Geology (UJA) specimen VF1: the 620 mm long holotype cervical V of A. philadelphiae. This specimen is argued by some authors to pertain to an animal with a c. 3 m long neck (Frey & Martill, 1996; Martill et al., 1998), a dimension which would make large azhdarchids among the longest-necked animals outside of Sauropoda (Taylor & Wedel, 2013) and Plesiosauria, despite their necks being formed of only nine vertebrae (Bennett, 2014).

However, recent discoveries of two proportionally short, isolated azhdarchid cervical vertebrae from the Maastrichtian Sebeş Formation (Transylvanian Basin) of western Romania have prompted suggestions that some azhdarchids may have been proportionally short necked (Vremir, 2010; Vremir et al., 2015). The first of these specimens, LPV (FGGUB) R.2395, was interpreted as a cervical IV from a small azhdarchid with an estimated 3 m wingspan (Vremir et al., 2015). The second represents a gigantic azhdarchid: Transylvanian Museum Society (Cluj-Napoca, Romania) specimen EME 315 (Fig. 1). This latter bone is proportionally short and wide, of robust construction and bears—for a pterosaur—remarkably thick bone walls. Details of bone structure and provenance led Vremir (2010) to suggest it may represent a cervical III from Hatzegopteryx, a giant azhdarchid described from the middle member of the Densuş-Ciula Formation, Maastrichtian of Vãlioara, northern Haţeg basin, deposits contemporary and adjacent to the Sebeş Formation. We discuss the taxonomic identity of the specimen further below. Vremir (2010) concluded that the size and shape of EME 315 is so distinct relative to that of other azhdarchids that it must reflect a departure from expected azhdarchid anatomy and lifestyle.

Figure 1 Giant azhdarchid cervical vertebra referred to Hatzegopteryx sp. (A–D) line drawings of EME 315 in anterior (A) right lateral (B) ventral (C) and dorsal (D) views; (E) proportions of EME 315 compared to other azhdarchid cervicals: note atypical combination of length/width ratio (l:w) and length compared to other azhdarchid cervicals, and especially against the only other known giant cervical, Arambourgiania (UJA RF1).

Light shading indicates damage; dark shading indicates filler. Abbreviations: co, cotyle; hy, hypapophysis; nc, neural canal; nsa; neural spine (anterior region); nsp, neural spine (posterior region); pnf, pneumatic foramen; prz, prezygapophysis; poz, postzygapophysis; vprzt, ventral prezygapophyseal tubercle (fused cervical rib). Scale bar is 100 mm.

The concept of short necked azhdarchids is yet to be explored in detail, despite the significance it has for our understanding of azhdarchid palaeoecology and disparity. The functional anatomy of the long, stiffened azhdarchid neck has been the most controversial element in discussions of azhdarchid lifestyles (e.g., Witton & Naish, 2015; Averianov, 2013, and references therein), so understanding its variation and biomechanics is paramount to advancing palaeobiological appreciation of the group. Here, we investigate the radical morphological differences between EME 315 and other azhdarchid cervicals from two angles. Firstly, we attempt to estimate the probable neck length of EME 315 and other azhdarchids (both giant and smaller species) to assess possible variation in their proportions and form. Secondly, we assess the bending strength of two giant azhdarchid vertebrae (EME 315 and UJA VF1) to appreciate variation in structural properties and functionality, and relate these to contemporary ideas of azhdarchid behaviour and ecology. It is imperative to these studies that we also investigate the likely identity and vertebral position of EME 315, and this is also discussed below.

Methods

Taxonomic and anatomical identity of EME 315

EME 315 possesses multiple apomorphies of azhdarchid pterosaur cervical vertebrae, including the characteristic ‘bifid’ neural spine, large, dorsoventrally flattened zygapophyses and a low centrum (e.g., Andres & Ji, 2008; Averianov, 2010; Buffetaut & Kuang, 2010; Vremir et al., 2013). It can thus be referred to Azhdarchidae with confidence. We agree with Vremir (2010) that comparable size, anatomy, and geographical and geological provenance all indicate affinities with Hatzegopteryx, a robust giant azhdarchid first described from nearby Vălioara in the Haţeg Basin (Buffetaut, Grigorescu & Csiki, 2002; Buffetaut, Grigorescu & Csiki, 2003). We draw specific attention to the ventral bone wall of EME 315: at 4–6 mm thick, it is considerably thicker than the 2.6 mm or less reported from most other giant azhdarchids (including the giant Arambourgiania holotype cervical—Frey & Martill, 1996; Martill et al., 1998) but is comparable to bone walls of the H. thambema holotype humerus (Laboratory of Vertebrate Palaeontology, Geological and Geophysical Faculty, University of Bucharest, Romania) FGGUB R1083 (Buffetaut, Grigorescu & Csiki, 2003). A large, elongate cervical vertebra from the Maastrichtian of the French Pyrenees was also described as having thick bone walls of 2–6 mm (Buffetaut et al., 1997) so it is possible that this feature was more widespread in azhdarchids. The spongiose internal texture visible at the broken end of EME 315 also recalls the aberrant internal structure of the skull and humerus of the H. thambema holotype (Buffetaut, Grigorescu & Csiki, 2002). We consider Hatzegopteryx and EME 315 to possess a bone construction atypical among pterosaurs, and a close relationship between these specimens likely. The Sebeş Basin material does not overlap with the H. thambema holotype, so we, accordingly, provisionally identify the Sebeş Basin vertebra as Hatzegopteryx sp. only. This referral of EME 315 to Hatzegopteryx is supported by the lack of firm evidence for a second giant azhdarchid in Romania, as well as the fact that, while multiple azhdarchid taxa are known to have been contemporaneous in several Late Cretaceous faunas (Vremir et al., 2013; Vremir et al., 2015), we have yet to discover evidence that more than one giant taxon inhabited a given fauna.

Isolated azhdarchid cervicals have typically been regarded as offering little insight to their position within the cervical series, except perhaps for cervical V, which appears distinctly elongate (Frey & Martill, 1996; Martill et al., 1998). Recent work on relatively complete azhdarchid cervical skeletons indicates that their vertebrae may show consistent characteristics specific to the position in the cervical series (Suberbiola et al., 2003 (sensu Kellner, 2010); (Averianov, 2010; Averianov, 2013) (Fig. 2). Work in this area must be regarded as provisional given that complete azhdarchid necks, or even sufficient material to completely reconstruct entire cervical series, remain few in number. However, we consider known azhdarchid necks of consistent enough form that the likely vertebral position of well-preserved azhdarchid cervicals, such as EME 315, can be determined with some degree of confidence.

Figure 2 Characteristics of azhdarchid vertebrae across their cervical series, demonstrated by several azhdarchid taxa.

(A) Azhdarcho lancicollis cervical III (ZIN PH 131/44), left lateral aspect; (B–C) Quetzalcoatlus sp. cervical III (TMM 41544.16) in dorsal (B) and left lateral (C) aspect; (D) A. lancicollis cervical IV (ZIN PH 144/44), left lateral aspect; (E) Q. sp. cervical V (TMM 41455.15), left lateral aspect; (F) Arambourgiania philadelphiae cervical V (UJA VF1), dorsal aspect; (G–H) A. lancicollis cervical VI (ZIN PH 147/44) in left lateral (G) and posterior (H) aspect (note especially large neural spine); (I) A. lancicollis cervical VII (ZIN PH 138/44), dorsal aspect; (J) Phosphatodraco cervical VII (OCP DEK/GE 111), left lateral aspect; (K) A. lancicollis cervical VIII (ZIN PH 137/44), dorsal aspect. Abbreviations as for Fig. 2, also with con; condyle; ex, exapophysis; ns, neural spine. (A, D, G–H) and (K) after Averianov (2010); (F) after Frey & Martill (1996); (J) after Suberbiola et al. (2003).

Vremir (2010) considered EME 315 as a cervical III, but we consider this unlikely. The neural spines of cervical III in Azhdarcho lancicollis (Zoological Institute of the Russian Academy of Sciences, St. Petersburg, Russia, ZIN PH 131/44) and Quetzalcoatlus sp. (Texas Memorial Museum, Austin, USA, TMM 41544.16) extend for the length of the entire centrum and lack any obvious reduction in height at mid-length (Fig. 2A; Howse, 1986; Averianov, 2010), a significant contrast to the bifid neural spine of EME 315. Indeed, Howse (1986) reported that the Quetzalcoatlus cervical III neural spine is at its highest point mid-way along its length, a marked contrast to the condition in EME 315. The proportions of cervical III cotyles, which are approximately twice as wide as tall and subequal in height to the neural arch, also contrast with EME 315, as does the continuous tapering of cervical III zygapophyses when viewed in dorsal aspect. Cervical IIIs also seem generally longer-bodied than the proportionally short EME 315. We find greater similarity with other azhdarchid cervicals (below) and thus disagree with a cervical III identity for EME 315.

Azhdarchid cervicals IV and V can be up to eight times longer than wide (Lawson, 1975; Howse, 1986; Frey & Martill, 1996). Their neural spines comprise low anterior and posterior ridges with a mid-length so reduced that they are confluent with the vertebral corpus, sometimes being represented by a faint, narrow ridge at best (Figs. 2D–2F). EME 315 is not elongate relative to its width (Fig. 1E) and, though possessing a bifid neural spine, the breadth of the preserved neural spine bases suggests they were robust, tall structures. Azhdarchid cervical VIs seem similar to fourth and fifth elements, but have a proportionally tall posterior neural spine (Figs. 2G–2H). EME 315 contrasts with most or all of these conditions, and thus likely pertains to a posterior section of the neck—that is, to cervicals VII or VIII.

Strong similarity occurs between EME 315 and cervicals VII and VIII of Azhdarcho lancicollis (ZIN PH 138/44 and 137/44, respectively (Averianov, 2010; Averianov, 2013), Figs. 2I–2K), with the most notable similarity pertaining to cervical VII. The cotyle heights of these vertebrae are characteristically shallower than their neural arches, and four times wider than high (Averianov, 2010). The cotyle width:height ratio of EME 315 approximates this at ca. 3.7. Both EME 315 and Azhdarcho cervical VII possess hypapophyses, a contrast to cervical VIII of Azhdarcho where a hypapophysis is absent (Averianov, 2010). Reconstructed length:width ratios of EME 315 and the posterior cervicals of Azhdarcho are similar (1.36 in Azhdarcho cervical VII, 1.06 in cervical VIII, versus 1.25 in EME 315; based on a reconstructed EME 315 length and width of 300 mm and 240 mm, respectively), as are the presences of pneumatic foramina dorsal to the neural canal. The relatively splayed prezygapophyses of cervicals VII and VIII in Azhdarcho also correspond well with EME 315, although they are much smaller in Azhdarcho cervical VIII. The articular faces in the latter are joined to the vertebral body via a constricted bony shaft, whereas the zygapophyses of EME 315 and Azhdarcho cervical VII are more massive overall. Cervical VII in Azhdarcho and EME 315 are also similar in having a tapered ‘waist’ mid-way along the length of the centrum. This feature is absent in cervical VIII of Azhdarcho which has, in contrast, subparallel lateral margins. The pneumatic foramina are larger than the neural canal in Azhdarcho’s cervical VII, which contrasts with the condition in EME 315 and cervical VIII of Azhdarcho. EME 315 also lacks pneumatic foramina on the lateral surface of the centrum, in contrast to Azhdarcho’s cervical VIII where they are present. The neural spines on the posterior cervicals of Azhdarcho are unknown, but those of the posteriormost cervicals of Phosphatodraco mauritanicus are proportionally tall and anteroposteriorly restricted (Fig. 2J; Suberbiola et al., 2003). This condition matches the one that appears to have been present in EME 315.

EME 315 thus possesses a combination of anatomical traits that are a good match for the posterior cervical vertebrae of at least two other azhdarchid taxa, and it differs markedly from the middle or anterior neck vertebrae of any taxon. We note particular similarity with cervical VII of Azhdarcho and hence provisionally consider a seventh cervical position most likely for EME 315, the caveat being that additional discoveries of azhdarchid posterior cervical vertebrae are needed to bolster our identification.

Size of the EME 315 individual

We refrain from providing a specific wingspan estimate for the EME 315 individual because the relationships between wingspans and cervical vertebrae are not reliably predicted using existing data. Disagreements over the wingspan of the individual represented by the Arambourgiania holotype cervical (stated as having a wingspan of 7–8 m wingspan by Suberbiola et al. (2003) and yet argued as 10 m or more by others—Frey & Martill, 1996; Steel et al., 1997; Martill et al., 1998) demonstrate the uncertainty surrounding size estimates of the largest pterosaurs known only from vertebral remains. Vremir (2010) indicated that the great width of EME 315 suggested a similarly expanded postcervical column and perhaps a much larger overall size than that of other giant azhdarchids. This interpretation is questionable as the cervical and anteriormost dorsal vertebrae of giant pterodactyloids are wider and more massive than the rest of the axial column (Bennett, 2001; Kellner et al., 2013). A lack of study on proportional scaling of the pterosaur axial column further precludes reliable predictions of the dimensions of the dorsal column belonging to the animal represented by EME 315.

Nevertheless, it is possible to provide a qualified assessment of the general size represented by this vertebra. EME 315 is the most robust pterosaur cervical yet reported and likely conforms to approximate size predictions for the Hatzegopteryx holotype, estimated to have a 10–12 m wingspan from the FGGUB R1083 humerus (Buffetaut, Grigorescu & Csiki, 2003; Witton & Habib, 2010). The size of pterodactyloid cervical condyles and cotyles appears to be relatively uniform along the cervical series (e.g., Anhanguera (Wellnhofer, 1991a); Quetzalcoatlus sp. (Witton & Naish, 2008); Azhdarcho (2010)), allowing us to assume that the 150 mm wide cotyle of EME 315 is similar to the condylar and cotylar dimensions present along the preceding part of the neck. In the reconstructed neck of Azhdarcho, and in completely known necks of Anhanguera, atlas cotyle width (assumed to correspond to the dimensions of the occipital condyle) is 30–40% of condyle and cotyle width in the remainder of the neck: the 55 mm wide occipital condyle of the H. thambema skull therefore corresponds to the 150 mm wide cotyle of EME 315. The unprecedented width and robust construction of EME 315 also corresponds with the unusually broad skull of H. thambema, estimated to span 500 mm across the quadrates (Buffetaut, Grigorescu & Csiki, 2003). We take these rough comparisons to indicate that EME 315 probably represents an animal at the upper known limit of pterosaur size.

Neck length estimate

Incredibly long necks incorporating elongate, tubular mid-cervical vertebrae are a well-known feature of Azhdarchidae (e.g., Nessov, 1984; Frey & Martill, 1996; Kellner, 2003; Unwin, 2003; Witton & Naish, 2008; Averianov, 2013). However, published attempts to estimate the length of giant azhdarchid necks are rare and presently limited to isometric scaling of Quetzalcoatlus bones to the same linear proportions as the Arambourgiania holotype (Frey & Martill, 1996; Steel et al., 1997). Subsequent discussions of neck length in giant azhdarchids (e.g., Martill, 1997; Taylor & Wedel, 2013) have relied on these figures. However, some pterosaur necks, like those of virtually all long-necked tetrapods, are known to scale with positive allometry against body size (Wellnhofer, 1970): this calls into question the assumption that azhdarchid necks scaled isometrically and the accuracy of these predicted values.

To estimate and compare the lengths of cervicals III–VII for the EME 315 individual and other azhdarchids, we compiled vertebral length data from six azhdarchid necks: four associated and complete cervical series—representing three skeletons of Zhejiangopterus linhaiensis (Cai & Wei, 1993), and the holotype of Phosphatodraco mauritanicus (Suberbiola et al., 2003, as interpreted by Kellner, 2010)—in addition to reconstructed, composite skeletons of Azhdarcho and Quetzalcoatlus sp. (Steel et al., 1997; Averianov, 2013) (Table 1). Our sample represents animals with wingspans ranging from 2.5–4.6 m and cervical III–VII lengths of 326–1495 mm. Regression analyses of these data provided reliable relationships between azhdarchid cervical vertebrae and cervical III–VII length (Fig. 3). Surprisingly, we find that azhdarchid necks scale rather differently to other long necked tetrapods. In most long necked animals—examples include sauropods, giraffes, plesiosaurs and tanystropheids (Tschanz, 1988; O’Keefe & Hiller, 2006; Parrish, 2006)—extreme neck length is often associated with a disproportionate increase in the size of cervical vertebrae (i.e., larger animals have disproportionately elongate neck bones against neck length). However, azhdarchid cervical vertebrae seem to largely retain their proportions with increasing neck length, scaling with only slight deviations from isometry. Cervicals III and VII show slightly negative scaling exponents of 0.88 and 0.78 (respectively), while cervicals IV–VI show exponents within 0.9–1.11 (Fig. 3). This ‘conservative’ approach to scaling is discussed more below.

Table 1 Azhdarchid cervical vertebrae data used in neck length estimates.

	Lengths (mm)	
Taxon	Zhejiangopterus linhaiensis	Phosphatodraco mauritanicus	Azhdarcho lancicollis	Quetzalcoatlus sp.	
Reference	Cai & Wei (1993)	Suberbiola et al. (2003), Kellner (2010)	Averianov (2013)	Steel et al. (1997)	
Specimen number	M1323	M13234	M1328	OCP/DEK GE 111	Reconstruction	Reconstruction	
Cervical number			
III	36	50	57	110	57.2	170.0	
IV	114	82	92	190	78.1	265.0	
V	142	84	98	225	156.2	410.0	
VI	120	72	81	190	102.3	380.0	
VII	90	38	56	150	60.0	270.0	
CIII–VII neck length	502	326	384	865	453.75	1495.00	
Proportion of CV/neck length	0.283	0.258	0.255	0.260	0.34	0.27	
Proportion of CVII/neck length	0.179	0.117	0.146	0.173	0.13	0.18	

Figure 3 Relationships between azhdarchid cervical vertebrae to cervical III–VII length.

Figure 4 Metrics and cross sections used in estimates of bending strength analysis.

(A) EME 315 in dorsal view showing line of modelled section (dotted line) and projected 300 mm length; (B) UJA VF1 in dorsal view showing line of section and projected 770 mm length (Frey & Martill, 1996); (C) cross section and dimensions of EME 315; (D) cross section of UJA VF1. Note difference in shape and bone wall thicknesses in (C) and (D).

Vertebral strength analysis

Azhdarchid cervicals are essentially hollow tubes with near-circular or elliptical cross sections (Fig. 4), and are thus conducive to beam loading calculations to ascertain their strength. We modelled bending load capacity for both UJA VF1 (the holotype vertebra of Arambourgiania) and EME 315 based on their minimal central diameters, and using both their preserved and estimated total lengths (Table 2). Cortical thicknesses were measured from broken mid-shaft sections of each bone. To enhance comparability between these vertebrae, we also modelled a hypothetical Hatzegopteryx cervical V based on length projections from our azhdarchid neck dataset and the centrum dimensions of EME 315: we estimate this bone’s length as 413 mm. This also provides a minimum estimate of neck strength because, as noted above, cervical V is the longest bone in the azhdarchid neck and thus the most susceptible to distortion under loading. Vertebral sections were modelled as consistent along the vertebral length and internal supporting structures were not factored into our equations. Because the vertebrae in question are elliptical in cross-section, we modelled both dorsoventral and lateral bending resistance. To calculate second moment of area (I, required to calculate section modulus for stress calculations) for each vertebral axis, we used: (1) I=π∕4R1R23−R 3R43

where R1 and R2 represents the total bone radii in perpendicular x and y axes (respective to loading regime), and R3 and R4 represent radii of the internal bone cavity. Bone stress was modelled using cantilever-style loading, where one end of the bone is fixed and the total length of the bone equals the moment arm. Stress values reflect those experienced at the supported end of the bone. Vertebrate bones are rarely loaded as true cantilevers in life but such a reductionist approach provides a quantified means of comparing bone structure and robustness (Witton & Habib, 2010). We calculated stresses (σ, Mpa) experienced at the supported end of the vertebrae during cantilevered loading: (2) σ=WL∕Z

where L is bone length (mm), W (N) is the weight loaded onto the bone and Z is section modulus (second moment of area/distance to neutral surface of the vertebra). Calculating bone strength requires some assumptions about the Young’s modulus of pterosaur bone. We follow Palmer & Dyke (2010) in using 22 GPa—a value agreeing with several avian long bones—which seems a reasonable proxy for pterosaur bones. Following Currey (2004) and Palmer & Dyke (2010), we used the relationship between Young’s modulus and yield stress in tension of 162 MPa. We modelled a range of values reflecting different upper limits for giant pterosaur body mass (180–250 kg) for W to demonstrate the sensitivity of our results and calculate Relative Failure Force (RFF; Witton & Habib, 2010) for each model. RFF is bone failure force, in bending, divided by total body weights. Although pterosaur axial elements were unlikely to ever bear a full loading of body mass in life, it provides a useful proxy by which we might compare the results here with those of other studies (e.g., Witton & Habib, 2010) and to compare strengths of pterosaur vertebra against their respective body masses.

Table 2 Giant azhdarchid cervical vertebra bending strength compared.

									Section modulus	Maximum stress (Mpa)	RFF	
Taxonomic ID	Specimen	Body mass (kg)	W (N)	Vertebral length (mm)	Centrum width radius (mm)	Centrum height radius (mm)	Cortical thickness (mm)	R/t	Sagittal bending	Coronal bending	Sagittal bending	Coronal bending	Sagittal bending	Coronal bending	
Hatzegopteryx sp.	EME 315	250	2,452	240	57.5	37	5	9.45	25307	32830	23.25	17.92	6.97	9.04	
Hatzegopteryx sp.	EME 315 (reconstructed)	250	2,452	300	57.5	37	5	9.45	25307	32830	29.06	22.40	5.57	7.23	
Hatzegopteryx sp.	EME 315 (hypothetical CV)	250	2,452	412.7	57.5	37	5	9.45	25307	32830	39.98	30.82	4.05	5.26	
Arambourgiania philadelphiae	UJA VF1	250	2,452	620	24	27.5	2.6	9.90	4819	4455	315.40	341.20	0.51	0.47	
Arambourgiania philadelphiae	UJA VF1 (reconstructed)	250	2,452	770	24	27.5	2.6	9.90	4819	4455	391.71	423.75	0.41	0.38	
Hatzegopteryx sp.	EME 315	200	1,961	240	57.5	37	5	9.45	25307	32830	18.60	14.34	8.71	11.30	
Hatzegopteryx sp.	EME 315 (reconstructed)	200	1,961	300	57.5	37	5	9.45	25307	32830	23.25	17.92	6.97	9.04	
Hatzegopteryx sp.	EME 315 (hypothetical CV)	200	1,961	412.7	57.5	37	5	9.45	25307	32830	31.98	24.66	5.07	6.57	
Arambourgiania philadelphiae	UJA VF1	200	1,961	620	24	27.5	2.6	9.90	4819	4455	252.32	272.96	0.64	0.59	
Arambourgiania philadelphiae	UJA VF1 (reconstructed)	200	1,961	770	24	27.5	2.6	9.90	4819	4455	313.36	339.00	0.52	0.48	
Hatzegopteryx sp.	EME 315	180	1,765	240	57.5	37	5	9.45	25307	32830	16.74	12.90	9.68	12.55	
Hatzegopteryx sp.	EME 315 (reconstructed)	180	1,765	300	57.5	37	5	9.45	25307	32830	20.93	16.13	7.74	10.04	
Hatzegopteryx sp.	EME 315 (hypothetical CV)	180	1,765	412.7	57.5	37	5	9.45	25307	32830	28.79	22.19	5.63	7.30	
Arambourgiania philadelphiae	UJA VF1	180	1,765	620	24	27.5	2.6	9.90	4819	4455	227.09	245.67	0.71	0.66	
Arambourgiania philadelphiae	UJA VF1 (reconstructed)	180	1,765	770	24	27.5	2.6	9.90	4819	4455	282.03	305.10	0.57	0.53	

Results and Discussion

Neck length of EME 315 and other azhdarchid pterosaurs

The results of our neck length estimates are summarised in Fig. 5. Our dataset shows a reasonable (r2 = 0.973) relationship between the length of cervical VII and the combined lengths of cervicals III–VII:

Figure 5 Measured and estimated azhdarchid pterosaur neck lengths against approximate wingspans.

(3) CIII−VII=17.283CVII0.7835

where CIII−VII represents the length of cervicals III–VII (mm), and CVII represents the length of cervical VII (mm). Assuming EME 315 is a seventh cervical, its preserved length (240 mm) predicts a cervical III–VII length of only 1,266 mm, while the estimated total length (300 mm) projects cervical III–VII values of 1,508 mm. These values must be considered low given the size of EME 315 and its indications of body size similar to that of the H. thambema holotype. Using the estimated 770 mm length (Frey & Martill, 1996), we modelled the cervical III–VII length of Arambourgiania at 2,652 mm, a value shorter than estimates based on strictly isometric scaling (2,817 mm; Steel et al., 1997) but still 75% longer than that predicted for the EME 315 azhdarchid. This discrepancy is further borne out in our estimate of 412 mm for a Hatzegopteryx cervical V—almost half the estimated length of Arambourgiania cervical V. Indeed, predicted cervical values of EME 315 match those measured from the reconstructed neck of the 4.6 m wingspan Quetzalcoatlus sp. (Steel et al., 1997): its estimated cervical V length and neck length are near identical to values measured from Q. sp. (410 mm and 1495 mm, respectively) despite this taxon being substantially smaller (Fig. 6).

Figure 6 Speculative skeletal reconstructions of Hatzegopteryx sp. and Arambourgiania philadelphiae (estimated wingspans ≥10 m—Frey & Martill, 1996; Buffetaut, Grigorescu & Csiki, 2003) to show discrepancy in neck length alongside a ‘typical’ azhdarchid body plan.

(A) Hatzegopteryx skeleton in lateral aspect; (B) dorsal view of EME 315 and FGGUB R1083 jaw elements, proportionate to actual size, suggesting Hatzegopteryx bore a wide, as well as relatively short, neck construction (soft-tissue outline in black). Jaw width after Buffetaut, Grigorescu & Csiki (2003); (C) reconstructed Arambourgiania philadelphiae cervicals III–VII in lateral aspect; (D) 4.6 m wingspan Q. sp. skeleton in lateral aspect; (E) Q. sp. cervical vertebrae III–V and skull in dorsal view; Note how the neck length of Hatzegopteryx is similar to this much smaller pterosaur. H. thambema holotype (FGGUB R1083) and undescribed referred elements are shown in (A); known elements of A. philadelphiae (UJA JF1) indicated in white shading in (C). Scale bar represents 1 m.

These calculations converge in establishing that Hatzegopteryx had a proportionally short neck (Vremir, 2010) c. 50–60% of the length expected for a ‘typical’ giant azhdarchid like Arambourgiania. Our estimates indicate that giant azhdarchids included both Hatzegopteryx-like forms with short, wide necks, and Arambourgiania-like species with long, gracile necks. The former befits an animal with the unusually robust cranial anatomy known for H. thambema and is consistent with the view that this pterosaur was robust overall (Buffetaut, Grigorescu & Csiki, 2002; Buffetaut, Grigorescu & Csiki, 2003). As noted above, short necks have been postulated for a much smaller Romanian azhdarchid known from a likely cervical IV, LPV (FGGUB) R.2395 (Vremir et al., 2015). This neck of this animal, considered to have a 3–4 m wingspan, was estimated at 352–419 mm using an earlier version of the data presented above: we revise this estimate upwards here to 460 mm. Nevertheless, this value is still shorter than that measured from smaller azhdarchids (e.g., the 2.5 m wingspan Zhejiangopterus linhaiensis, 502 mm measured neck length) and suggests that short necks may not be restricted to giant taxa (Vremir et al., 2015). Overall, these data suggest that there is more variation in neck proportions and robustness within Azhdarchidae than previously anticipated: the concept of the clade as one with a uniformly long-necked morphotype (e.g., Witton & Naish, 2008) now warrants significant reappraisal.

Neck biomechanics in giant azhdarchids

EME 315 represents an anatomical extreme among pterosaur neck vertebrae: its size, bone wall thickness and massiveness are unprecedented among other flying reptile remains. Its structural properties, and utility within a possibly shorter variant of the azhdarchid neck, are therefore significant not only to our understanding of azhdarchid palaeobiology as a whole, but in that they represent a hitherto unreported morphological class of pterosaur anatomy.

Our strength analysis (Table 2) shows that Hatzegopteryx neck vertebrae are considerably stronger than those of Arambourgiania. Even at the lowest loading threshold, and in its strongest bending plane (sagittal), the holotype Arambourgiania cervical does not withstand the strain of one bodyweight. At most, the UJA VF1 vertebrae has RFFs of 0.57 (1,765 N loading in sagittal plane), this decreasing to 0.38 in 2,452 N coronal loading. Hatzegopteryx, however, shows consistent capacity for the withstanding of high stresses. The (reconstructed) 300 mm long EME 315 model has an RFF of 10.04 when loaded with 1,765 N in the coronal pane, and maintains high RFFs (5.57) even when loaded by 2,452 N on its weakest axis. The longer (412 mm) hypothetical Hatzegopteryx cervical IV is also consistently strong in all tests, able to withstand 4.05–7.3 RFFs in various loading regimes.

These findings confirm predictions that giant azhdarchid vertebrae are not functionally uniform (Vremir, 2010). The detailed anatomy of giant azhdarchid cervicals provide insights into the contrasting figures generated by our bone strength analysis. Arambourgiania cervical V can be viewed as a giant variant on a ‘typical’ azhdarchid cervical, being a thin-walled (bone wall thickness 2.6 mm), elongate tube supported internally by a network of bony trabeculae (Frey & Martill, 1996; Martill et al., 1998). It mainly differs from other azhdarchid cervicals in bearing a mid-centrum section which is taller than wide (55 mm tall vs. 48 mm wide). As is well documented for other long pterosaur bones, this form is ideally suited to maximising stiffness, and thus resisting bending and torsion over long dimensions and within constrained loading regimes. The ratio of bone shaft thickness to wall thickness (bone radius/bone thickness, R/t) in UJA VF1 is 9.9, a value greater than recorded from other tetrapods but comparable to those measured from large pterosaur wing bones (Currey, 2002; Fastnacht, 2005).

Frey & Martill (1996) suggested that the unusually tall cross section of Arambourgiania likely improved its resistance to dorsoventral loading, and this is corroborated by our bending analysis. Dorsoventral expansion of a cervical vertebra is an efficient means to increase vertical bending strength without incurring additional mass (Frey & Martill, 1996), and we might predict this to be an evolutionary response to an increase in the weight of the neck and head in large azhdarchid pterosaurs. Even accounting for the ‘conservative’ scaling of pterosaur necks (Fig. 3), mass compounds exponentially against length, and giant pterosaurs would thus have experienced proportionally higher loading on their neck skeleton than similarly proportioned smaller species. We predict that the size and weight of large azhdarchid heads explains their relatively ‘conservative’ cervical scaling compared to long-necked species with small heads, as the heads of the latter impart relatively low structural demands on the supporting neck and permit development of proportionally longer neck components even at large size. In having heads which are predicted to be several metres long (Buffetaut, Grigorescu & Csiki, 2003; Witton, 2013), giant azhdarchids would have experienced much greater structural demand on their cervical skeletons, even accounting for cranial pneumaticity, and this almost certainly lessened the potential for pronounced cervical allometry.

As with most pterosaur bones, the greatest risk of structural failure to UJA VF1 is buckling: this can be caused by high compressive loads along the long axis of the vertebra or large bending moments. This may explain why the R/t of the Arambourgiania cervical is not as high as those measured from other long pterosaur bones (Fastnacht, 2005 reports an R/t of 20 for some pterosaur bones), as lowering R/t is one way to increase buckling strength.

The structural characteristics of EME 315 frequently contrast with this configuration. As noted above, the vertebra is proportionally short overall, and although its mid-centrum section has an elliptical shape typical for an azhdarchid, it is broader than other azhdarchid centra in all respects, being 74 mm tall by 115 mm wide. The large second moment of area created by the expanded centrum can be seen as being particularly significant as goes resisting bending through experimental modelling of a vertebra with the Hatzegopteryx section profile and the 770 mm length predicted for Arambourgiania cervical V. Even when loaded at 2,452 N, this hypothetical vertebra still produces high (over 2.17) RFF scores. By contrast, the smaller, thinner-walled section of Arambourgiania only achieves an RFF of 1.47 when shortened to 300 mm (the predicted complete length of EME 315) and modelled with the lightest loading in our experiments.

The EME 315 bone wall is relatively thick (4-6 mm) which means that—despite the size of the centrum—it has an R/t comparable to that of Arambourgiania at 9.45. This is noteworthy, as its larger size hypothetically permits a much higher R/t, which would be advantageous to decreasing mass and increasing performance against bending (see Currey, 2002 for discussion). However, it may be that the thicker cortices of this bone enhanced buckling strength without drastically altering bending strength (Currey, 2002) or that its cross-sectional proportions are sufficient to provide high bending resistance alone. Such thick bone walls are not without precedent in pterosaurs—they appear in certain dsungaripterid limb bones (Fastnacht, 2005), a partial vertebra from another European azhdarchid (Buffetaut et al., 1997) and the Hatzegopteryx type material (Buffetaut, Grigorescu & Csiki, 2002; Buffetaut, Grigorescu & Csiki, 2003). Buckling resistance, as well as increased performance against compressive loading, has been posited as an explanation for this phenomenon (Fastnacht, 2005).

Several other structural features are of interest for EME 315. Well-preserved endosteal regions of EME 315 show that a system of camellate bone, rather than the trabeculae seen in Arambourgiania (Martill et al., 1998), occupied at least the ventral part of the centrum’s interior. Such tissues seem pervasive throughout Hatzegopteryx bones, also being present in the jaw and humerus. We interpret these features as evidencing further resistance to buckling elsewhere in the skeleton. Furthermore, the already large mid-length centrum of EME 315 is considerably expanded at its anterior and posterior ends. This allows for broadened cotyle/condyle articulations and a greater capacity to distribute high stresses between vertebral joints, and their relatively wide, shallow profile is ideally shaped to resist torsion.

Assuming that the general characteristics and proportions of these giant azhdarchid neck vertebrae apply to their entire cervical series (which seems reasonable, given the profiles of other pterosaur vertebrae), we predict that at least two structural configurations existed among these giant forms. Selection pressures on Arambourgiania seem to have prioritised mass reduction and stiffness, which are ideal for elongating bones at the expense of loading capacity. We predict that the anterior cervical skeleton and crania of Arambourgiania were relatively slender and lightweight, with a gracile skull more akin to that of Quetzalcoatlus than the proportionally broad or deep skulls of Hatzegopteryx or the unnamed Texas Memorial Museum specimen 42489-2. EME 315 seems contrarily adapted: its cross-sectional proportions, massive features and thick bone walls are not advantageous for producing a long, lightweight neck skeleton (at least within the context of pterosaur anatomy), but better suited to resisting high bending and compressive stresses. Assuming the other neck bones of the EME 315 individual were similarly adapted, Hatzegopteryx must have possessed a significantly stronger neck skeleton than Arambourgiania, and perhaps the strongest neck of any known pterosaur. Our stress analysis accords with observations that the very large jaw bones of Hatzegopteryx indicate a very wide (0.5 m), and thus potentially relatively large and heavy, skull (Buffetaut, Grigorescu & Csiki, 2002; Buffetaut, Grigorescu & Csiki, 2003).

Supporting and utilising the azhdarchid neck skeleton

The robustness and apparent strength of EME 315 raises questions about the function of the Hatzegopteryx neck, particularly with respect to how it may have performed in tasks other than supporting a large skull. Investigating this requires some appreciation of pterosaur neck musculature. Pterosaur cervical myology has not featured prominently in technical discussions of this group, but artistic representations of azhdarchids—many of them overseen by pterosaur researchers—frequently show an extremely reduced cervical musculature relative to the typical tetrapod condition. We assume that these reconstructions were produced following observation of mid-series vertebrae, which are very long, have reduced processes, and have indications of limited arthrological range (Averianov, 2013).

However, azhdarchid fossils—including the specimens discussed here—show that the assumption of a paltry, reduced neck musculature represents an oversimplification and is inconsistent with anatomical data from other animals. Our arguments can be summarised as follows: (1) azhdarchid skeletal anatomy suggests that certain muscle groups related to neck function were indeed minimised, but that many aspects of axial, skull and pectoral skeletal anatomy show potential for large muscle attachments; (2) comparisons made between azhdarchid neck skeletons and those of extant animals suggest they are not as atypical as often assumed, and that reptilian cervical musculature correlates well with large muscle attachment sites on azhdarchid cervicals; and (3), that various aspects of azhdarchid anatomy counter proposals of a reduced degree of soft-tissue neck support. We will briefly explore these points here to further elaborate on the functional capacity of giant azhdarchid necks.

Our most general observation is that complete, associated azhdarchid neck skeletons show that they are not solely composed of simple, stiff-jointed, near-featureless tubes. As outlined in Fig. 2, cervicals III, VI, VII and (probably) VIII possess relatively prominent neural spines, indicating differential development of epaxial musculature (Witton & Naish, 2008). The ‘tubular’ morphology often ascribed to their neck skeletons only really applies to cervicals IV and V. Averianov (2013) demonstrated that intervertebral cervical articulations are variable along the neck, those of the posterior vertebrae being less restrictive than those of the anterior- and mid-sections. In these respects, azhdarchid necks are comparable to those of other amniotes. X-rays of living animals show that the middle section of the cervical series is often relatively immobile, and that the majority of movement in the neck is achieved via movement at either end of the cervical series (Vidal, Graf & Berthoz, 1986; Graf, De Waele & Vidal, 1992; Graf, De Waele & Vidal, 1995; Taylor, Wedel & Naish, 2009). Relatively long-necked mammals (examples include horses, deer, giraffes and camels), as well as extinct long-necked reptiles such as tanystropheids, possess reduced processes and relative immobility associated with their mid-length cervical vertebrae (Fig. 7; Goldfinger, 2004; Renesto, 2005). Azhdarchid neck skeletons are thus typical in that greater complexity and robustness was present at the extreme ends of their cervical skeleton, as well as in neighbouring cranial or torso skeletal elements; this was surely associated with the anchoring of powerful neck musculature and large ligaments at the base and anterior end of the neck. These are optimal positions from which to support and operate long necks. In view of this, the elongate and tubular, relatively immobile mid-series vertebrae of azhdarchids should be viewed as a pronounced development of a skeletal adaptation common across tetrapods, not as an unusual or unprecedented anatomical configuration.

Figure 7 Azhdarchid craniocervical skeleton compared to those of some other tetrapods.

(A) Tanystropheus cf. longobardicus; (B) reconstruction of Zhejiangopterus linhaiensis cervical skeleton, vertebral morphology adapted from Averianov (2010); (C) Giraffa camelopardalis; (D) Camelus dromedarius; (E) Odocoileus virginianus. Note that the mid-series vertebrae of all taxa—even those with highly complex, strongly-muscled neck skeletons—have reduced features compared to those at the posterior and anterior: the fact that azhdarchid mid-series cervicals have reduced features does not necessarily reflect underdeveloped cervical soft-tissues. (A) reconstructed from fossils illustrated by Rieppel et al. (2010); (B) reconstructed from Cai & Wei (1993) and Averianov (2010); (C–E) after Goldfinger (2004). Images not to scale.

Azhdarchid skeletons show ample attachment sites for neck musculature. For example, the occiput of Hatzegopteryx shows obvious signs of substantial soft-tissue attachment: the nuchal line is well developed and long, and its dorsolateral edges are deeply dished and marked with vertical scarring (Buffetaut, Grigorescu & Csiki, 2002; Buffetaut, Grigorescu & Csiki, 2003). Comparison with extant reptile anatomy Herrel & De Vree, 1999; Cleuren & De Vree, 2000; Tsuihiji, 2005; Tsuihiji, 2010; Snively & Russell, 2007; Snively et al., 2014 suggests that these features reflect large insertion areas for transversospinalis musculature (specifically m. transversospinalis capiti and the m. epistropheo-capitis/splenius group), cervical musculature devoted to head and neck extension and lateral flexion. The large neural spines on posterior azhdarchid cervicals and anterior thoracic vertebrae provide potential origin sites for m. transversospinalis capiti, while the long neural spine of cervical III likely anchored m. epistropheo-capitis. The opisthotic process of Hatzegopteryx is poorly known but was evidently large and robust and likely facilitated attachment of large neck extensors and lateral flexors (m. semispinalis capitis/spinocapitis posticus). Similarly, the broken basioccipital tuberosities of Hatzegopteryx are long even as preserved: neck and head flexors anchoring to these (m. longissimus capitis profundus, m. rectus capitisventralis) would have had high mechanical advantage. The length and size of these occipital features suggest that large muscles with augmented lever arms were anchored to the azhdarchid skull. Witmer et al. (2003) and Habib & Godfrey (2010) made similar observations about the occipital regions of other pterodactyloids: at least the anterior neck skeleton of pterosaurs was likely strongly muscled.

At the other extreme of the axial column, the azhdarchid scapulocoracoid suggests that their superficial neck musculature may have been well developed. Their scapulae are large and dorsoventrally expanded compared to those of other pterosaurs (e.g., Elgin & Frey, 2011), permitting broad insertions of m. levator scapulae and m. serratus (Bennett (2003) shows their likely origin in other pterosaurs). These muscles originate on the anterior cervicals in modern reptiles and can function as neck elevators and retractors if the scapulae are immobile. Azhdarchid scapulocoracoids articulated tightly with the dorsal vertebrae and sternum (Frey, Buchy & Martill, 2003) and were buried within deep flight musculature, so were likely capable of little, if any, motion. Contraction of cervical-pectoral muscle groups would thus likely elevate the neck, and asymmetric contraction of these muscles would move the neck laterally. These muscles (or homologues thereof) are particularly large in long-necked, large-headed mammals such as horses and deer (Goldfinger, 2004), and we propose that the enlarged pectoral skeleton of azhdarchids may indicate similar enhancement of the posterior neck musculature.

Comparison with the anatomy of modern reptiles suggest that both m. levator scapulae and m. serratus, as well as muscles operating within the cervical series, anchored to the lateral faces of the neural arch and zygapophyses in azhdarchids (Herrel & De Vree, 1999; Cleuren & De Vree, 2000; Snively & Russell, 2007). This is important for consideration of azhdarchid palaeobiology, it being a clear indication that neural spine height is not the only indicator of neck muscle size. Reptilian cervical extensor musculature, such as m. longissimus cervicis and m. transversospinalis cervicis, originate and insert on cervical zygapophyses as well as the vertebral corpus (Herrel & De Vree, 1999; Snively & Russell, 2007). Other muscles, including those superficial muscles outlined above and m. longus colli ventralis, originate on cervical ribs, diapophyses and transverse processes (Herrel & De Vree, 1999; Cleuren & De Vree, 2000; Snively & Russell, 2007). These structures are reduced in azhdarchids, but not absent. Juvenile specimens show that vestigial cervical ribs occur on the ventral surfaces of their prezygapophyses (Godfrey & Currie, 2005), fusing to the zygapophyses in older animals to form the ventral face of the prezygapophysis (Unwin, 2003). In well preserved specimens, fused cervical ribs form prominent ventral prezygapophyseal tubercles (Company, Ruiz-Omeñaca & Pereda Suberbiola, 1999; Vremir et al., 2015). The retention of cervical ribs in tubercle form may indicate that these structures maintained a functional role, perhaps persisting as attachment sites for muscles ancestrally anchored to non-reduced cervical ribs, diapophyses or transverse processes. We refrain from making more specific comment on this issue until an improved understanding of pterosaur cervical musculature is achieved, but note that well-preserved azhdarchid zygapophyses have complicated morphologies with crests, prominences, concave facets and well-defined edges (e.g., Frey & Martill, 1996; Company, Ruiz-Omeñaca & Pereda Suberbiola, 1999; Averianov, 2010; Vremir et al., 2013; Vremir et al., 2015): anatomy expected of structures that act as anchorage sites for prominent musculature. The atypically elongate, broad zygapophyses of azhdarchids (e.g., Howse, 1986; Unwin, 2003; Kellner, 2003; Witton & Naish, 2008) can be viewed with new significance if, as proposed here, they accommodated muscle attachment.

Finally, the idea that azhdarchids had thinly muscled necks is at odds with their cranial proportions, which are among the most extreme of any animal. The skulls of azhdarchids are proportionally huge (Fig. 6; Cai & Wei, 1993; Kellner & Langston Jr, 1996; Witton, 2013) and—even accounting for their high degree of pneumaticity—would have subjected their neck tissues to high amounts of strain and stress. A well-developed system of ligaments and epaxial musculature was likely needed for cranial movement and support.

We thus propose that hypotheses of highly reduced neck muscles in azhdarchids are likely erroneous. The reduction of some axial structures—in particular the neural spines of the mid-series cervicals and small size of cervical rib homologues—suggest that some muscle groups were likely reduced, but other areas for muscle attachment were prominent enough to indicate that their necks were neither weak nor underpowered. Indeed, several of their likely attachment sites must be viewed as expanded compared to those of other pterosaurs, and with effective mechanical advantage for operating the head and neck.

Our hypotheses regarding azhdarchid neck musculature allow us to make some provisional, general comments on the vertebral myology of giant forms. We note that areas likely to anchor muscle—such as neural spines and zygapophyses—of EME 315 are proportionally expanded. The bifid neural spine of EME 315 is broken at the base of each process, but the broken surfaces are sufficiently broad and elongate (Fig. 1) to suggest that the spines were broad, long and perhaps tall when complete. The geometry of the zygapophyses are complex. Low crests and prominent edges extend from the vertebral corpus towards their articular surfaces, and their lateral and medial faces show complex concavities and edges: we posit that these mark muscle scarring. The ventrolateral surfaces of the EME 315 corpus are also notably concave and meet the ventral face along a defined, sweeping edge. These features suggest that EME 315 was well-muscled in life. This seems appropriate given the size of the Hatzegopteryx skull, and those features indicating large muscle insertions on its occipital face.

The holotype cervical of Arambourgiania may also show some evidence of muscle scarring: a sagittal crest on its anterior ventral surface and two low crests on the dorsal surface of the prezygapophyses. These latter features are topographically similar, though less defined, to crests seen on EME 315 and other azhdarchid vertebrae. However, the overall potential area for muscle attachment in this giant vertebra is much lower than it is in EME 315. The broken section of the anterior surface of the neural spine is smaller than that seen in EME 315, indicating a shallower neural spine overall. The zygapophyses are also shorter and more gracile. These differences might be partly explained by the different likely positions of EME 315 and UJA VF1 within the cervical skeleton (a cervical V is expected to have lesser muscle attachment than preceding or following vertebrae) but better known azhdarchid necks suggest that generalities of morphology will be common in other, adjacent vertebrae along the column (Fig. 5). We therefore conclude that Arambourgiania likely had a relatively lightly muscled neck relative to that of Hatzegopteryx. This is in keeping with the reduced strength of UJA VF1 predicted in our testing.

Disparity and ecological diversity in giant azhdarchids

EME 315 and the other Hatzegopteryx material provides the strongest evidence yet that azhdarchids were not anatomically uniform (Vremir et al., 2013; Witton, 2013). Understanding the overall form of azhdarchids is hampered by a lack of associated material, but fragmentary specimens indicate that azhdarchids were variable in at least three major anatomical respects (Figs. 5 and 8). The first is neck type, since some taxa had relatively short (though perhaps not shorter than expected for other pterodactyloids), robust necks (such as Hatzegopteryx; R2395), and others had much longer, more gracile and mechanically weaker necks (e.g., Quetzalcoatlus sp., Arambourgiania). The second is cranial morphotype: this also comprises robust forms, with relatively short skulls and proportionally broad jaws (e.g., the possible azhdarchid Bakonydraco; Javelina Formation specimen TMM 42489-2), and gracile forms with elongate rostra and slender jaws (Quetzalcoatlus sp.; Zhejiangopterus; Alanqa). Some azhdarchids also appear to have relatively slender rostra, as indicated by the concave dorsal skull margin of Azhdarcho (Fig. 8A, Averianov, 2010). A third category concerns the wing skeletons: we note that the relatively abbreviated metacarpal IV and proximal wing phalanx of the diminutive azhdarchid Montanazhdarcho minor contrasts markedly with the elongate distal forelimb elements of Quetzalcoatlus sp. and Zhejiangopterus (McGowen et al., 2002). It has been speculated that azhdarchids might be roughly grouped into ‘robust’ and ‘gracile’ forms based on these differences (Witton, 2013). It certainly seems appropriate to consider forms like Hatzegopteryx ‘robust’ and others—e.g., Quetzalcoatlus and Zhejiangopterus—‘gracile’, but some taxa show ‘mixed’ anatomies (e.g., Montanazhdarcho has proportionally stocky wing bones, but elongate neck bones (McGowen et al., 2002)), suggesting these categories must be considered loose. Azhdarchid body plans may have been rather more varied than imagined previously.

Figure 8 Azhdarchid disparity in cranial and limb anatomy.

(A) ZIN PH 112/44, rostral fragment of Azhdarcho lancicollis showing concave dorsal skull margin (after Averianov, 2010); (B) anterior skull and mandible of TMM 42489-2, unnamed azhdarchid from the Javelina Formation, USA; (C) restored skull of Quetzalcoatlus sp. (based on Kellner & Langston Jr, 1996); (D) skull of Zhejiangopterus linhaiensis (based on Cai & Wei, 1993); (E) MOR 69I, Montanazhdarcho minor holotype pectoral girdle and left forelimb (note stunted metacarpal IV); (F) M1323 postcrania of Z. linhaiensis. Abbreviations: car, carpals; cer, cervical vertebrae; cor, coracoid; fem, femur; hum, humerus; mcIV, metacarpal IV; pt, pteroid; rad, radius; tib, tibia; ul, ulna; wpI, wing phalanx I. Scale bars represent 100 mm, except for A (10 mm).

Figure 9 Diversity in predicted life appearance and ecologies for giant azhdarchid pterosaurs.

(A) two giant, long-necked azhdarchids—the Maastrichtian species Arambourgiania philadelphiae—argue over a small theropod; (B) the similarly sized but more powerful Maastrichtian, Transylvanian giant azhdarchid pterosaur Hatzegopteryx sp. preys on the rhabdodontid iguanodontian Zalmoxes. Because large predatory theropods are unknown on Late Cretaceous Haţeg Island, giant azhdarchids may have played a key role as terrestrial predators in this community.

Our assessment of vertebral mechanics in Hatzegopteryx and Arambourgiania suggests that azhdarchid necks had drastically different functional capabilities. We presume that cranial and cervical disparity reflects distinct foraging habits and prey preferences, with robust azhdarchids tackling relatively larger prey than their gracile counterparts. The stout, thick-walled cervicals of Hatzegopteryx, as well as its generally reinforced bones and wide jaws (Buffetaut, Grigorescu & Csiki, 2002; Buffetaut, Grigorescu & Csiki, 2003), seem better suited to tackling larger, more powerful prey, or for using greater force and violence when obtaining food, than azhdarchid species with thin-walled bones, long, gracile necks and narrow skulls. Undescribed fossils likely referable to Hatzegopteryx (including additional skull and limb elements that cannot be described here) show that robust construction was consistent across its body. The high resistance to bending stresses and indications of large cervical muscles in Hatzegopteryx are consistent with this concept, as are the inverse findings for Arambourgiania.

Modern studies on azhdarchid foraging behaviour suggest that they were terrestrially-foraging generalists (Witton & Naish, 2008; Witton & Naish, 2015; Carroll, Poust & Varricchio, 2013; Witton, in press). What little is known of giant azhdarchid anatomy is similar enough to that of the smaller, better known azhdarchids to assume that they also foraged terrestrially, albeit perhaps with a greater emphasis on carnivory. We propose that gracile giants like Arambourgiania consumed relatively small prey such as early juvenile and hatchling dinosaurs, large eggs and other diminutive components of Cretaceous terrestrial ecosystems (Fig. 9A). This is in keeping with proposals that some giants occupied ‘middle tier’ predatory niches in some Cretaceous ecosystems (Witton & Naish, 2015). Hatzegopteryx, however, shows potential for tackling much larger prey items, perhaps even killing animals too large to ingest whole (modern azhdarchid analogues, such as storks, are capable of attacking large animals, and killing human children, with their azhdarchid-like beaks: see Witton & Naish (2015) for discussion). Hatzegopteryx is the largest terrestrial predator known in Maastrichtian eastern Europe by some margin (Witton & Naish, 2015): its size, robust anatomy, and the deficit of other large carnivores in well-sampled European deposits implies that it may have been an arch predator in its community (Fig. 9B). The idea that a pterosaur may have played such an important role in a terrestrial Cretaceous ecosystem is far removed from previous interpretations of azhdarchids and pterosaurs generally, and perhaps a clear sign of how far pterosaur studies have progressed in recent decades (see Wellnhofer, 1991b; Hone, 2012; Witton, 2013 for overviews of pterosaur research).

Finally, the growing evidence for distinct bauplans within Azhdarchidae complicates assessments of pterosaur disparity at the close of the Cretaceous and ideas surrounding the extinction of the group. Azhdarchids dominate pterosaur faunas in the Maastrichtian, with only two localities recording non-azhdarchid pterosaurs from this time (Price, 1953; Longrich, Martill & Andres, 2016). Assumptions that azhdarchids were morphologically uniform have led to proposals that Maastrichtian pterosaurs were ecologically constrained at the end of the Cretaceous, and that their extinction represents the unspectacular end of a long, gradual pterosaurian decline (Unwin, 2005; Witton, 2013). The identification of clear distinctions in form and function within Azhdarchidae, along with the recent potential identification of the first small-bodied azhdarchid species from Campanian sediments (Martin-Silverstone et al., 2016), indicates that latest Cretaceous pterosaurs were not as ecologically homogenous as previously thought, and that their extinction may have coincided with their exploitation of niches previously unused in pterosaur evolution. Pterosaur extinction in the K/Pg event may thus have been more significant than traditionally considered.

This work reflects discussion with a number of individuals on azhdarchid pterosaur anatomy, animal scaling and functional anatomy. We are grateful to S Brusatte, E Buffetaut, G Dyke, M Habib, D Hone, A Kellner, C Palmer, M Norell, D Unwin, M Vremir and M Wedel for their assistance in helping us develop the ideas outlined in this paper. We are grateful to three referees (Eric Snively, David Hone and Elizabeth Martin-Silverstone) who provided very useful and constructive comments on an earlier version of this paper.

Additional Information and Declarations

Competing Interests

Author Contributions

Data Availability

The authors declare there are no competing interests.

Darren Naish and Mark P. Witton conceived and designed the experiments, performed the experiments, analyzed the data, contributed reagents/materials/analysis tools, wrote the paper, prepared figures and/or tables, reviewed drafts of the paper.

The following information was supplied regarding data availability:

Raw data is available in Tables 1 and 2.

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
