# Peer review of "Neck biomechanics indicate that giant Transylvanian azhdarchid pterosaurs were short-necked arch predators"

_PeerJ, doi:10.7717/peerj.2908_

## Round 0.1 · original submission · Minor Revisions

All of the reviewers found the manuscript to be generally sound and well-structured. Each flagged some areas for improvement. I've considered their suggestions and all of them seem apt and reasonable.

I only have one comment of my own to add. Since you are considering the disparity in azhdarchid body forms, and in particular the cervical vertebrae, you may want to at least briefly reference BMR P2002.2, an extremely elongate azhdarchid cervical from Hell Creek, described by Henderson and Peterson in the March 2006 issue of JVP. I believe that it would have had a higher length/width ratio than any of the vertebrae currently plotted in Figure 1E. You're not required to discuss that specimen, but it is an interesting counterpoint from the opposite end of the proportional spectrum.

Please address all of the points raised by the reviewers in your revised manuscript or in your rebuttal letter. I'll look forward to seeing an even better version of this work soon.

·

Basic reporting

In general I think this has been done very well. The one thing I am surprised has not been mentioned is any of the medium-sized azhdarchid material from Alberta, even just as an introduction to where azhdarchids come from and their diversity. There is a pretty well preserved partial skeleton with both a nice cervical vertebra and humerus that could be mentioned.

Experimental design

The only comment I have here is that it is not made clear how the authors calculated the cortical thickness (or internal radius) for the specimens that seemed to be complete in the middle of the shaft. Additionally, cortical thickness can vary in a single section, so has it been averaged? If the thickness ranged from 2-6mm in the vertebra, why was 5mm used as the thickness at this point? And how did they determine the thickness at this point?

Validity of the findings

Conclusions appear to be well supported by both the literature and the data presented here. No major concerns with the findings or interpretations.

Additional comments

In general, I'm very happy and excited about this paper. I think that it presents some very interested and novel information about azhdarchids, further supporting the idea that we don't fully understand them. I have added comments of specific things to an annotated PDF, but I have 3 general concerns/comments to point out:
1. Make sure that you are consistent and correct with the terminology of stress/strength/bending/etc. In engineering terms they all have very specific definitions and can mean different things if you use the wrong one. General strength is not the same as bending strength or stress. And stress itself can refer to several types (e.g. compression, tension and torsion).
2. I feel that other medium-large size azhdarchids (e.g. the ones in Alberta and Quetzalcoatlus sp.) should at least be mentioned somewhere as further examples of azhdarchids and diversity just in terms of completeness of background info. The Alberta material especially has a very nice cervical vertebra and humerus that could be compared to this material as a comparison of some smaller animals. Just an idea, but not major.
3. Not a big thing at all, but I'm hesitant in the identification of Hatzegopteryx for this vertebra. As it's stated several times how weird it is, and there is no overlapping material with it and what is identified as Hatzegopteryx, it seems weird to then say that Hatzegopteryx has a neck unlike other pterosaurs found so far. Why could it not be something different? I'm conservative with this kind of thing though, so it isn't a major concern, just a comment.

·

Basic reporting

NA

Experimental design

I am not qualified to comment on the calculations made for bone wall strength etc. but they certainly seem appropriate and the documentation and description of the materials used is adequate.

Validity of the findings

NA

Additional comments

This is a simple and straightforwards paper (not that this is a criticism) and there are no major issues with this. I have only a few relatively simple suggestions that will improve the flow of the manuscript and add a few small details to clarify some points. Please see the attached document for comments and areas to look at.

·

Basic reporting

The writing and approach are clear. I try not to impose on style too much, but in a few cases you can tighten up the language ("by the fact that"= "because"; "we note that" is unnecessary).

A few suggested additions to literature on neck muscles. There's been a lot of action in the last decade or so. However, no need to change the text or interpretation, except for one or two suggested additions.

Experimental design

There are some subtle but important corrections to structural terminology. I like the indexing of structural property results to body mass.

Define what's being compared allometrically to what (minor point).

Validity of the findings

Mechanically valid approach, results, and interpretations.

Additional comments

The research has exciting and evocative implications for azhdarchid behavior. Shoebills and marabou storks have deep lower jaws compared with the skeletal and life restorations of Hatzegopteryx. Were they really that slender? Perhaps hulk them out a bit.

---

## Round 0.2 · accepted · Accept

Thank you for your diligence in addressing the concerns of the reviewers. I am satisfied with the revised manuscript, and I am happy to accept it for publication in PeerJ.

The decision of whether or not to publish the peer reviews alongside the paper is entirely yours, and will not affect how your paper is handled going forward. However, I encourage you to do so. This is a great example of strong manuscript being made even stronger by a constructive review process. More importantly, all three reviewers chose to sign their reviews, and making the reviews public allows the reviewers to receive more credit for their efforts, and also contributes to the emerging culture of fairness and transparency in editing and peer review.